# HDAC6 inhibitor ACY-1083 shows lung epithelial protective features in COPD

Jenny Horndahl[1], Rebecka Svärd[1], Pia Berntsson[1], Cecilia Wingren[1], Jingjing Li[2], Suado M. Abdillahi[1], Baishakhi Ghosh[3], Erin Capodanno[4], Justin Chan[5], Lena Ripa[6], Annika Åstrand[7], Venkataramana K. Sidhaye[8], Mia Collins[1] *

1 Bioscience COPD/IPF, Research and Early Development, Respiratory & Immunology, BioPharmaceuticals R&D, AstraZeneca, Gothenburg, Sweden, 2 Bioscience Asthma, Research and Early Development, Respiratory & Immunology, BioPharmaceuticals R&D, AstraZeneca, Cambridge, United Kingdom, 3 Department of Environmental Health and Engineering, Johns Hopkins Bloomberg School of Public Health, Johns Hopkins University, Baltimore, Maryland, United States of America, 4 Department of Biology, Krieger School of Arts & Sciences, Johns Hopkins University, Baltimore, Maryland, United States of America, 5 Department of Public Health Studies, Krieger School of Arts & Sciences, Johns Hopkins University, Baltimore, Maryland, United States of America, 6 Medicinal Chemistry, Research and Early Development, Respiratory & Immunology, BioPharmaceuticals R&D, AstraZeneca, Gothenburg, Sweden, 7 Project Leader Department, Research and Early Development, Respiratory & Immunology, BioPharmaceuticals R&D, AstraZeneca, Gothenburg, Sweden, 8 Division of Pulmonary and Critical Care Medicine, Johns Hopkins School of Medicine, Johns Hopkins University, Baltimore, Maryland, United States of America

* Mia.collins@astrazeneca.com

**Data Availability Statement:** All relevant data are within the manuscript and its Supporting Information files.

**Funding:** Research reported in this publication was supported by the National Heart, Lung, and Blood

## Abstract

Airway epithelial damage is a common feature in respiratory diseases such as COPD and has been suggested to drive inflammation and progression of disease. These features manifest as remodeling and destruction of lung epithelial characteristics including loss of small airways which contributes to chronic airway inflammation. Histone deacetylase 6 (HDAC6) has been shown to play a role in epithelial function and dysregulation, such as in cilia disassembly, epithelial to mesenchymal transition (EMT) and oxidative stress responses, and has been implicated in several diseases. We thus used ACY-1083, an inhibitor with high selectivity for HDAC6, and characterized its effects on epithelial function including epithelial disruption, cytokine production, remodeling, mucociliary clearance and cell characteristics. Primary lung epithelial air-liquid interface cultures from COPD patients were used and the impacts of TNF, TGF-β, cigarette smoke and bacterial challenges on epithelial function in the presence and absence of ACY-1083 were tested. Each challenge increased the permeability of the epithelial barrier whilst ACY-1083 blocked this effect and even decreased permeability in the absence of challenge. TNF was also shown to increase production of cytokines and mucins, with ACY-1083 reducing the effect. We observed that COPD-relevant stimulations created damage to the epithelium as seen on immunohistochemistry sections and that treatment with ACY-1083 maintained an intact cell layer and preserved mucociliary function. Interestingly, there was no direct effect on ciliary beat frequency or tight junction proteins indicating other mechanisms for the protected epithelium. In summary, ACY-1083 shows protection of the respiratory epithelium during COPD-relevant challenges which indicates a future potential to restore epithelial structure and function to halt disease progression in clinical practice.

Institute (NHLBI R01HL124099 &
HLR01HL151107, VKS), and the Ludwig Family
Department of Medicine Physician-Scientist Grant
(VKS). The funders had no role in study design,
data collection and analysis, decision to publish, or
preparation of the manuscript.

**Competing interests:** I have read the journal's
policy and the authors of this manuscript have the
following competing interests: AstraZeneca
employees may hold shares in the company. This
does not alter our adherence to PLOS ONE policies
on sharing data and materials.

## Introduction

The human lungs are lined by a layer of epithelium which serves as a barrier to prevent direct access of inhaled luminal contents to the subepithelium. With each inhalation, the lung epithelium is in contact with the external environment, with the potential of exposure to harmful environmental particles and infectious organisms [1]. The respiratory epithelium constitutes a physical barrier between the outer and inner environments, and the epithelium dictates the initial responses to these stimuli. One mechanism by which the epithelium does this is by tightly regulating transport across the epithelial barrier by means of cell-cell junctions. These include tight junctions (TJs), adherens junctions (AJs), gap junctions, and desmosomes, which all allow for epithelial cells to be connected to their neighbors and are the primary components of the physical barrier formed by the epithelium [2–4]. In addition to regulating the ability of particles to cross through the epithelium, the epithelial junctions also act to segregate the basal compartment from the apical compartment to create epithelial polarization [5, 6].

In recent years the lung epithelium has gained more interest for respiratory diseases such as COPD, asthma and IPF. These diseases are characterized by remodeling and destruction of the lung, resulting in chronic inflammation, susceptibility to infection and loss of small airways. COPD is currently the 4th leading cause of death in the world and accounts for 6% of all deaths globally (as more than 3 million died of COPD in 2012) [7–9]. Even early in disease, patients have changes in their small airways that can be observed by different technologies [10, 11]. During disease progression, infiltration of immune cells and remodeling, such as fibrotic lesions and emphysema, occur. Increased levels of inflammatory cytokines eg TNF, IL-6 and IL-1β, and several chemokines and proteases, such as MMPs, are detected systemically and/or in the lung at diagnosis [12–15]. Despite efforts with anti-inflammatory therapies, there is a huge unmet medical need for new ways of treating respiratory patients and to understand early events leading to inflammation and remodeling [16–18]. Recent papers [11, 19] have discussed the importance of epithelial integrity in lung diseases and reviewed the literature for evidence about epithelial injury as a symptom of damaged epithelium. Strikingly, the epithelial changes often appear before the onset of clinical symptoms such as changes in FEV1 and can be measured in patients using new and more sensitive imaging technologies, eg computed tomography (CT) [20, 21]. Other technologies for assessment of small airway function are emerging, eg oscillometry, enable both an early diagnosis and a way to document efficacy of an intervention [22, 23]. Cigarette smoke, which is a risk factor for COPD, and other irritants have been reported to disrupt TJs and AJs proteins, leading to a dysfunctional epithelium and a disrupted barrier which affects important functions like cilia length and beat frequency, mucus production and the normal organization of the epithelium [4, 24–28]. Mechanisms for this are discussed by Aghapour et al., [19] and there are also studies showing evidence of loss of these epithelial structures in COPD patients [24, 25, 29]. In fact, remodeling of the lung often starts with loss of TJ proteins altering the phenotype of cells resulting in epithelial plasticity with significant change in both the structure and function of the cells [30]. This starts a cascade of events with further remodeling down the line and loss of normal lung functions as a result.

HDAC6 is a histone deacetylase that has been described to play a role in epithelial and endothelial destruction and remodeling. The exact mechanisms for the described effects are to date unknown but its reported functions involve regulation of the cilia cell cycle and autophagy, and has substrates and interacting proteins such as ERK, PKCα, β-catenin, NF-κB, EGFR and peroxiredoxins, that have been implicated in epithelial dysfunction [31, 32]. HDAC6 activity is increased by several stimuli including cigarette smoke and TNF, and is described to regulate cilia disassembly, epithelial to mesenchymal transition (EMT), AJ

markers and oxidative stress, both *in vitro* and *in vivo* with similar results [33–40]. However, most studies performed to dissect the actions of HDAC6 use inhibitors such as Tubastatin A, Tubacin or Ricolinostat at high concentrations where the impact of other 'off targets' and/or other HDACs cannot be disregarded [41–44]. ACY-1083, a small molecule inhibitor highly potent for HDAC6 with a good selective profile, have been reported to have effect in injury-related *in vivo* models [45–49].

In this paper, we studied the effect of ACY-1083 on epithelial barrier function in primary COPD cells. We show protective features of this inhibitor in different *in vitro* injury models, where treated cell cultures display reduced damage and cytokine production, and sustained mucus movement resulting in a functioning and intact epithelium.

## Material and methods

### Characterization of inhibitors

**Inhibitors.** ACY-1083 (CAS1708113-43-2, Mw 348.35) is commercially available but was prepared according to a published procedure [50, 51]. Tubastatin A (HY-13271, Mw 371.86 HCl salt) and Ricolinostat (HY-16026, Mw 433.5) were obtained from MedChemExpress.

**HDAC enzymatic activity.** Assays were performed at Eurofins Pharma Discovery Services Catalog reference HDAC1 2491; HDAC2 2492; HDAC3 2083; HDAC4 2493; HDAC5 2494; HDAC6 2495; HDAC7 2610; HDAC8 2247; HDAC9 2611; HDAC10 2662 and HDAC11 2663.

**Plasma protein binding.** Plasma protein binding analysis was performed as reported previously using human blood plasma samples [52].

**Pharmacokinetic analysis in cell media.** To determine the stability and binding to plastic and cell media components, cell media was added to a test tube and spiked with ACY-1083 to a final concentration of 10 μM. Compound concentration was then analysed after 18hrs incubation at 37°C by LC-MSMS (Method DMPK&BAC 001–04).

### Secondary pharmacology

Second pharmacology screening was performed at Eurofins Pharma Discovery Services in a panel of 190 *in vitro* radioligand binding and enzyme assays covering a diverse set of enzymes, receptors, ion channels, and transporters. The cardiovascular panel (hERG, Nav1.5, Kv4.3 and IKs) were performed on the SyncroPatch 384PE (Nanion Technologies) high throughput patch clamp platform (Chinese hamster ovary K1 cell line) at room temperature in a 6 point cumulative assay.

### Air-Liquid interface culturing of human bronchial epithelial cells

**Patient characteristics from donor cells.** Primary COPD human bronchial epithelial cells (HBEC) were purchased from Lonza (Basel, Switzerland). We have used cells from four COPD donors and the information on these donors is limited but include age (years), gender (F/M) and smoking history (yes/no): donor 1 (Cat#00195275, Batch#0000630389, 54, F, yes), donor 2 (Cat#00195275S, Batch 0000370751, 65, F, yes), donor 3 (Cat#00195275S, Batch 0000436083, 59, M, yes), donor 4 (Cat#00195275S, Batch 0000430905, 66, M, yes). Additional five donors of primary healthy HBECs were purchased from Lonza and Epithelix SàRL (Geneva, Switzerland): donor 1 (Cat# hAECB, Batch# AB068001, 71, F, no), donor 2 (CC-2540, Batch# 0000501935, 56, M, yes), donor 3 (CC-2540S, Batch# 0000619260, 65, F, yes), donor 4 (hAECB, Batch# AB079301, 62, M, no) and donor 5 (hAECB, Batch# #AB077201, 63, M, no).

**Culturing conditions.** Passage one cells were seeded into T75 cell culture flasks (250,000 cells/flask) and expanded in PneumaCult ExPlus medium (StemCell Technologies, Vancouver, Canada) at 37˚C, 5% $CO_2$. Once cells reached 70% confluency, they were dissociated with Try-pLE Express dissociation media (Thermo Fisher Scientific, Waltham, USA) and frozen at -150˚C in ExPlus with 10% DMSO. Passage two cells were thawed and seeded (20,000 cells/membrane) into 24-well HTS Transwell plates (Corning, Wiesbaden, Germany) for air-liquid interface (ALI) culturing. Cells were expanded in ExPlus medium until >50% confluent (four days). Apical medium was then removed, and basolateral medium was replaced with Pneuma-Cult ALI medium (StemCell Technologies). Medium was changed three times per week during a four-week long differentiation phase.

For cilia beating frequency (CBF) and cell velocity measurements, donor cells of primary healthy HBECs were amplified on rat tail collagen I coated flasks and seeded on 12-mm-diameter Transwell inserts [4, 53, 54].

**Assessment of epithelial integrity.** To determine epithelial integrity of ALI cell cultures the trans-epithelial electrical resistance (TEER) of HBEC ALI cultures was measured using EVOM2 resistance meter (World Precision Instruments Inc, Sarasota, USA). For experiments conducted with healthy donors for the CBF and cell velocity the bronchial epithelial barrier function was evaluated by quantifying TEER using an EVOM epithelial voltohmmeter (World Precision Instruments, FL, USA) connected with the STX2 electrodes.

The paracellular permeability was determined by using fluorescein isothiocyanate labeled 4kD-dextran (Sigma-Aldrich, St. Louis, USA). 18 hours after addition of 0.2 mg FITC-dextran in 200 μL medium to the apical side of the ALI cultures, the fluorescence in the basolateral medium was measured using PHERAstar FSX (excitation 485 nm and emission 520 nm) (BMG LABTECH, Ortenberg, Germany). The time point was chosen for optimal window in the assays as well as practical reasons and has been kept consistent in between different stimulations throughout this work. Data was plotted as fold-change of 0.1% DMSO control.

**Challenge models in ALI.** All HBEC cultures had been in ALI for four weeks at the start of each experiment. All compounds and stimulus, except for whole cigarette smoke, were added on the basolateral side of the ALI cultures.

For experiments using TNF (R&D Systems, Minneapolis, Canada), TNF (ranging from 5–50 ng/ml) was added together with ACY-1083 or vehicle (0.1% DMSO) on the basolateral side at every medium change for up to ten days in four COPD donors. TEER and permeability were measured after seven days.

For TGF-β1 (R&D Systems) experiments, cells from two COPD donors were pre-treated with ACY-1083 or vehicle (0.1% DMSO) for three days before addition of TGF-β1 (0.4–10 ng/ml). Permeability was measured after 48 hours.

Cigarette smoke extract (CSE) experiments were performed using CSE containing media. In short, smoke from five filterless Kentucky research cigarettes, 3R4F (Kentucky Tobacco Research and Development Center, Lexington, USA), was bubbled through 25 ml of PBS at a speed of five minutes per cigarette. The CSE was filtered through a 0.2 μm sterile filter and stored at -80˚C until addition to the basolateral side of HBEC ALI cultures from two COPD donors. Three days of ACY-1083/0.1% DMSO pre-treatment was followed by 48 hours CSE-challenge. Permeability was measured after 48 hours.

Whole cigarette smoke (WCS) experiments were performed using the Smoking Robot VC 10Ⓡ S-TYPE (Vitrocell, Waldkirch, Germany). The Vitrocell smoke robot delivers either WCS or humidified air to the apical surface of the cell layers. Briefly, inserts with fully differentiated HBEC in ALI from one COPD donor were placed in the Vitrocell smoke chamber filled with DMEM medium (Thermo Fisher Scientific) and CS or humidified air was puffed onto the apical surface. ISO Puff Standard was used with each CS exposure consisting of two 3R4F

Kentucky research cigarettes (one 35 ml puff every 60 seconds). Cells were exposed to WCS four times during two days with a minimum of two hours elapsed between smoke exposures. Treatment with ACY-1083 started after the first smoke exposure. Permeability was measured 20 hours after the last smoke session.

Bacterial infection was carried out on fully differentiated HBEC ALI cultures from one COPD donor at three different occasions. Cells were pretreated with 10 μM ACY-1083 or 0.1% DMSO for nine days. *Haemophilus influenzae*, Pittman 576, type b (Culture Collection University of Gothenburg, Gothenburg, Sweden) was grown over night in Brain-heart infusion media (VWR, Radnor, PA, USA) at 37˚C. When optical density (620 nm) was above one, bacteria were centrifuged at 4000 rcf for ten minutes and then washed in cold PBS once. Serial dilutions of bacteria were done in PBS starting at 2e8 CFU/ml. 50 μL of bacterial dilutions were added to the apical side of the ALI cultures. After two hours incubation at 37˚C, apical surfaces were washed three times with PBS and paracellular permeability was measured 18hrs later.

**Alcian Blue/Periodic Acid-Schiff staining.** Eight days post-treatment, HBEC ALI-cultures were fixed in 4% paraformaldehyde for fifteen minutes and washed three times in PBS. Dehydration in ethanol and xylene followed by paraffin (Merck, New Jersey, USA) infiltration was done with a short program for biopsies (1 hour 55 minutes) on a Microm STP 120 Spin Tissue Processor (Thermo Fisher Scientific). Cell layers were embedded in paraffin and 4 μm thick sections were made using a Leica RM2165 microtome (Leica, Wetzlar, Germany). Alcian Blue/Periodic Acid-Schiff (AB/PAS) with Haematoxylin staining were performed using standard protocols on a Leica ST5020. After staining, the sections were dehydrated in ethanol and xylene, mounted with Pertex mounting medium (Histolab, Göteborg, Sweden) and scanned with a Aperio Scanscope slide scanner (Leica Biosystems, Buffalo Grove, USA).

**Quantification of cellular features from IHC sections.** IHC images were analyzed in HALO v3.1 (Indica Labs). For all analyses, images were annotated manually to exclude out of focus and damaged areas, and a random forest classifier was applied to intact areas of the sections to detect epithelial areas. All following analyses were done in epithelial areas only. For goblet cell counting, algorithm CytoNuclear 2.0.9 was used to segment cells based on nuclear staining and categorize them as AB/PAS (mucin) positive and negative cells based on cytoplasmic AB staining. Goblet cell counting was represented as the percentage of AB positive cells of total cells.

To measure epithelial thickness, the apical and basal sides of the epithelium in the sections were manually delineated using a pen annotation tool. Distance between paired apical and basal annotations was measured every 10 μm along the epithelium, and recorded in sequence as epithelial thickness.

To measure intraepithelial and intercellular AB staining, algorithm Area Quantification v2.1.7 was used. Two thresholds were applied to capture AB staining: a lower one to detect all AB staining in both goblet cells and intercellular areas, and a higher one to detect the stronger AB staining inside goblet cells only. Intercellular AB area was calculated by subtracting goblet AB staining (higher threshold) from total AB staining (by lower threshold).

**Occludin staining.** Eight days post-treatment, HBEC ALI-cultures were fixed in 4% paraformaldehyde for fifteen minutes and washed three times in PBS. Cells were permeabilized with 0.2% Triton X-100 in PBS for two hours and then incubated in 10% goat serum in PBS. Blocking buffer was replaced after one hour with 7 μg/mL mouse anti-occludin antibody (cat# 33–1500, Invitrogen) in PBS with 5% goat serum. After overnight incubation at 4˚C, cells were washed three times in PBS and incubated with 2 μg/mL goat anti-mouse IgG Alexa Fluor 594 (Invitrogen) for one hour at room temperature. Cells were washed in PBS and nuclei stained with 2 μM Hoechst 33342 (Thermo Scientific) for 30 minutes followed by three PBS washes.

Membranes were then cut from the inserts with a scalpel and mounted on glass slides in Pro-Long Gold Antifade Mountant (Thermo Fisher Scientific). Stained cell cultures were imaged using a Zeiss LSM 880 confocal microscope.

**Western blot of membrane protein from ALI cell cultures.** For extraction of membrane and cytosol proteins the cells were lysed according to the Mem-PER Plus Membrane Protein Kit protocol (#89842, ThermoFisher Scientific). The supernatants were kept frozen in −80˚C until total protein quantification. Quantification of total protein in the lysates was done using Pierce BCA Protein Assay Kit (#23227, Thermo Scientific) following the manufacturer´s instructions. Lysates containing 8 μg total protein were mixed with NuPAGE LDS Sample buffer 4x (NP007, Invitrogen), NuPAGE Sample Reducing Agent 10x (NP004, Invitrogen) and deionized water and then heated for ten minutes at 70˚C. The samples were loaded onto NuPAGE 4–12% Bis-Tris Gels (NP0335, Invitrogen) and run at 120V for 100 minutes in NuPAGE MOPS SDS Running buffer (NP0001, Invitrogen). The proteins were transferred to nitrocellulose membranes (LC2001, Invitrogen) at 35 mA overnight using NuPAGE Transfer buffer (NP0006, Invitrogen) supplemented with 20% methanol. The membranes were stained with Revert Total Protein Stain (926–11011, LI-COR Biosciences) for five minutes, washed and then imaged in the 700 nm channel using Odyssey CLX imaging system (LI-COR Biosciences). To correct for variation in loading the total signal intensity of all proteins in each lane were used for normalization. An inter-membrane control sample was loaded to all gels to assure there were no differences due to methodology between the blots. The stain was removed with Revert Destaining Solution (926–11013, LI-COR Biosciences) and the membranes were blocked in Intercept (TBS) Blocking Buffer (927–60001, LI-COR Biosciences) for one hour on a shaker. After blocking the membranes were incubated cold overnight with primary antibodies diluted in Intercept (TBS) Blocking Buffer (E-cadherin mAb #14472 Cell Signaling Technology 1:1000, β-catenin mAb #8480 Cell Signaling Technology 1:1000, Occludin mAb #33–1500 Invitrogen 1:500). On the next day the membranes were washed three times for ten minutes in Tris-buffered saline + 0.05% Tween (#91414, Merck) followed by incubation with IRDye Goat anti-Mouse 800CW (#926–32210, LI-COR Biosciences) and Donkey anti-Rabbit 680RD (#926–68073, LI-COR Biosciences) secondary antibodies (1:10000 dilution) for one hour at room temperature. After a final wash, three times for ten minutes in TBS-T, the fluorescent signals were detected using the Odyssey CLX imaging system and the signals were analysed using Image Studio software (v4.0, LI-COR Biosciences).

**Mucociliary clearance.** In the TNF (20 ng/ml) challenged cultures, the apical surfaces were washed with pre-warmed PBS the day before addition of 50 μL of CountBright Absolute Counting beads (Invitrogen, Carlsbad, CA, USA) diluted 1:10 000 in PBS with $Mg^{2+}$ and $Ca^{2+}$. Inserts were transferred to a 12-well glass bottom plate (Cellvis, Mountain View, USA) and monitored in a Zeiss LSM 880 (Carl Zeiss AG, Oberkochen, Germany) at 37˚C. Three regions in each well of each condition in three COPD donors were filmed and analyzed. Beads were tracked in the ImageJ software with the Fiji plugin Tracking [55] and the mean velocity of each bead was plotted.

**Ciliary beat frequency.** CBF was quantified for the differentiated healthy HBECs treated with vehicle, TNF, and TNF + ACY-1083. The plates containing the pseudostratified epithelia were incubated at 37 ˚C with 5% $CO_2$ in the 3i Marianis/Yokogawa Spinning-Disk Confocal microscope (Leica Microsystems, Sugar land, USA) as reported in our previous publications [53, 56, 57]. High-speed time-lapse videos were taken at 32X air at 100 Hz with a total of 250 frames using a scientific Hamamatsu C11440-42U30 CMOS camera (Bridgewater, USA). Five areas were imaged per insert. A Matlab (R2020a) script (validated against SAVA) (previously described in [58] was used to determine average CBF per video, to generate a heat map indicating CBF.

**Cell velocity.**   Cell migration was quantified by performing Particle Image Velocimetry (PIVlab) on Matlab using multi-pass cross-correlation analysis with decreasing interrogation window size on image pairs to obtain the spatial velocity field as described previously [59]. Using a phase contrast microscopy of 3i Marianis Spinning-Disk Confocal microscope (Leica Microsystems) at 32X air objective, time-lapse videos of the epithelial cell monolayer were captured for every five minutes for two hours following treatment with TNF, and/or ACY-1083, and the average velocity for the area was computed.

## Cytokine release

To assess cytokines released, basolateral supernatants were collected on day seven and immediately frozen at -80˚C until MSD analysis of IL-6, CCL2 and CXCL10. In short, U-PLEX MSD plates (Mesoscale Discovery, Rockville, Maryland, USA) were coated, washed and stored overnight. Standard curves were prepared and added in duplicates on each plate. Supernatants were thawed on ice and added to the plates, which were incubated on an orbital shaker for one hour before sulfo-tagged antibodies were added and incubated for another hour. Plates were washed, read buffer added and read on MESO SECTOR S 600 (Mesoscale Discovery, Rockville, Maryland, USA). Data was plotted as fold change to DMSO control.

## RNA analysis

To study transcriptional changes, RNA purification was carried out using the RNeasy Plus 96 kit (74192) (Qiagen, Venlo, The Netherlands) according to manufacturer's protocol. The liquid was collected and frozen at –80˚C until further analysis. RNA was quantified and cDNA synthesis was performed according to standard procedures using High-capacity cDNA Reverse Transcription Kit (4368813) (ThermoFisher, Waltham, MA, US). Real-time polymerase chain reaction (qPCR) was done using validated TaqMan® Gene Expression Assays (ThermoFisher, Waltham, MA, US), *IL8* (Hs00174103_m1), *CCL2* (Hs00234140_m1), *CXCL10* (Hs00171042_m1), *MMP9* (Hs00957562_m1), *IL1B* (Hs01555410_m1), *MUC5AC* (Hs01365616_m1), *MUC5B* (Hs00861595_m1) with TaqMan Fast Advanced Master Mix (4444557) (Applied Biosystems, Foster City, CA, US). Samples were run in triplicates and analyzed on the QuantStudio™ 7 Flex Real-Time PCR 384-well System (ThermoFisher Waltham, MA, US). Data was normalized to the housekeeping genes GAPDH (Hs99999905_m1), and RPLP0 (Hs99999902_m1) and delta-delta CT values were plotted (fold change to DMSO).

## Statistical evaluation

Statistical differences between samples were assessed with two-tailed paired t-test, one-way or two-way analyses of variance (ANOVA). Differences at p-values below 0.05 are considered significant. All statistical analyses were performed using Graphpad PRISM 8.4.2 (GraphPad Software, Inc., La Jolla, CA). All statistics comparing data within a treatment group was made using two-way ANOVA with Sidak's post test, and all comparisons between groups were made using two-way ANOVA with Dunnet's post-test. Tukey's test of variance was used for the thickness variation data analysed with R [60].

## Results

### ACY-1083 is highly selective against other HDACs

When using inhibitors as tools to validate a target, it is important to understand the selectivity towards other structurally similar targets. In the HDAC assay panel, HDAC6 inhibitors that are termed to be selective, such as Ricolinostat and Tubastatin A, have selectivity margins to

class I HDACs of as low as ten-fold and even lower towards Class II HDACs. Results from using these compounds in preclinical settings, claiming to be HDAC6-driven, should thus be interpreted with caution. ACY-1083 is a highly selective inhibitor of HDAC6 with a selectivity profile demonstrating >400-fold selectivity towards HDAC5 and HDAC7, >1000-fold selectivity over HDAC1, 4, 8, 9, 10 and 11, and >7000-fold selectivity over HDAC2 and 3 (Table 1). Structures can be found in S1 Fig. ACY-1083 is stable in human plasma with a free fraction of 23%. A simple spike test of adding 10 μM ACY-1083 to ALI cell media in a test tube showed an approximate 4 μM drug concentration after 18 hrs. This indicates that more than half of the compound is lost in binding to the tube/cell media proteins/components. In the preclinical models used in these experiments, this indicates that at concentration ranges up to 10 μM, ACY-1083 should target mainly HDAC6 and no other HDACs (but possibly 5 and 7 at below their respective $IC_{50}$). ACY-1083 has a reported cell potency ($IC_{50}$) of 30–100 nM, measured as the inhibition of deacetylation of α-tubulin [49]. ACY-1083 further displayed excellent selectivity in a panel of >190 *in vitro* radioligand binding and enzyme assays covering a diverse set of enzymes, receptors, ion channels, and transporters. The only activity identified below 30 μM was cyclooxygenase 2 (COX2, $IC_{50}$ = 6.48 μM), dopamine active transporter (DAT, $IC_{50}$ = 0.37 μM) and Sigma-1 receptor ($IC_{50}$ = 17 μM). Additionally, ACY-1083 had no activity below 30 μM in a panel of cardiovascular ion channels.

## ACY-1083 reduces paracellular permeability

To study epithelial barrier dysfunction in COPD, we used a 3D model system with primary human bronchial epithelial cells isolated from COPD donors. Cells were cultured at ALI until a pseudostratified epithelium was developed, consisting of ciliated cells, goblet cells and basal cells. These cells form tight junctions *in vitro* which will give rise to resistance over the membrane. Paracellular permeability of small polysaccharide molecules, such as FITC-dextran, is a way of measuring barrier integrity and has been used to capture changes in the epithelium.

ACY-1083 was added basolaterally to fully differentiated HBEC ALI cultures from four COPD donors at 10 μM for eight days and permeability was measured to assess barrier integrity. Passage of FITC-dextran molecules were significantly reduced with ACY-1083 in all four

**Table 1. HDAC enzyme activity for ACY-1083, Ricolinostat and Tubastatin A.**

| Compound | | ACY-1083* | Ricolinostat** | Tubastatin A** |
|---|---|---|---|---|
| | | $IC_{50}$ μM | | |
| Class I | HDAC1 | 17.2 | 0.515 | 1.91 |
| | HDAC2 | 70.7 | 1.78 | 9.86 |
| | HDAC3 | >87.5 | 0.954 | 12.4 |
| | HDAC8 | 15.8 | 0.826 | 0.345 |
| Class IIA | HDAC4 | 12.5 | 7.13 | 0.910 |
| | HDAC5 | 4.17 | 2.81 | 0.821 |
| | HDAC7 | 6.51 | 2.32 | 0.229 |
| | HDAC9 | 19.3 | 62.2 | 0.416 |
| Class IIB | HDAC6 | <0.01 | 0.0215 | 0.064 |
| | HDAC10 | 18.5 | 0.901 | 3.22 |
| Class IV | HDAC11 | 18.2 | 2.30 | 2.67 |

Demonstrating μM potencies for the different subtypes, color-coded for margins to HDAC6 (<100 fold = red, 100–1000 fold = yellow, >1000 fold = green),

*mean of two separate measurements

**one measurement

donors (p = 0.0023) (Fig 1), which indicates that ACY-1083 strengthens the epithelial barrier *per se* since no challenge was added.

## ACY-1083 reduced epithelial injury after multiple challenges

To mimic the epithelial injury seen in COPD, we developed a TNF-challenge model where cells were treated with TNF in the basolateral media, until a noticeable drop in resistance, and an increase in paracellular permeability was observed. 18hrs were chosen as time point where we saw a clear window in permeability. HBEC ALI cultures from four COPD donors were treated with 10 μM of ACY-1083 for eight days at the same time as TNF was added at different concentrations. After seven days, TEER and permeability were measured and showed that compound-treated wells had higher TEER at all concentrations of TNF (0, 5, 20 and 50 ng/mL), as well as lower permeability at 5, 20 and 50 ng/mL TNF as compared to the DMSO control (Fig 2A and 2B). ACY-1083 hence rescued the cells from TNF-induced epithelial barrier disruption.

In this experiment it was obvious that there is variation in donor susceptibility to TNF challenge but TEER and permeability results were always in accordance. Thus, we simplified the readouts to only measure permeability in the experiments with additional challenges.

Since COPD is a disease with multiple causes to epithelial injury, we also wanted to investigate other insults such as CSE, TGF-β, direct smoke and bacterial infection. In a similar fashion, CSE, TGF-β, whole smoke and *H. Influenzae* were added in different concentrations to COPD ALI cell cultures, and paracellular permeability was measured (S2 Fig). Treatment with

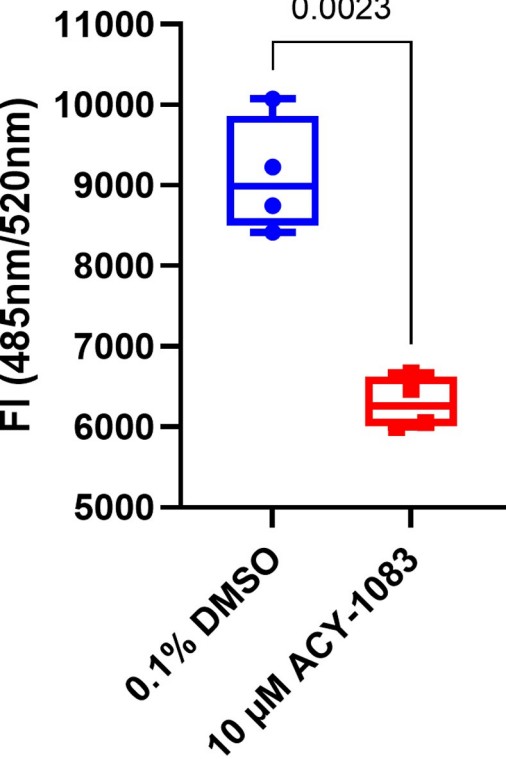

**Fig 1. ACY-1083 reduces paracellular permeability in unchallenged fully differentiated COPD HBEC ALI cultures.** Graph shows the fluorescence intensity of FITC-labelled dextran (4kDa) passed through cell layers to the basolateral side. Box plot with dots representing the average of three individual replicates for each of 4 COPD donors and whiskers showing min and max. Statistical analysis was performed using two-tailed paired t-test.

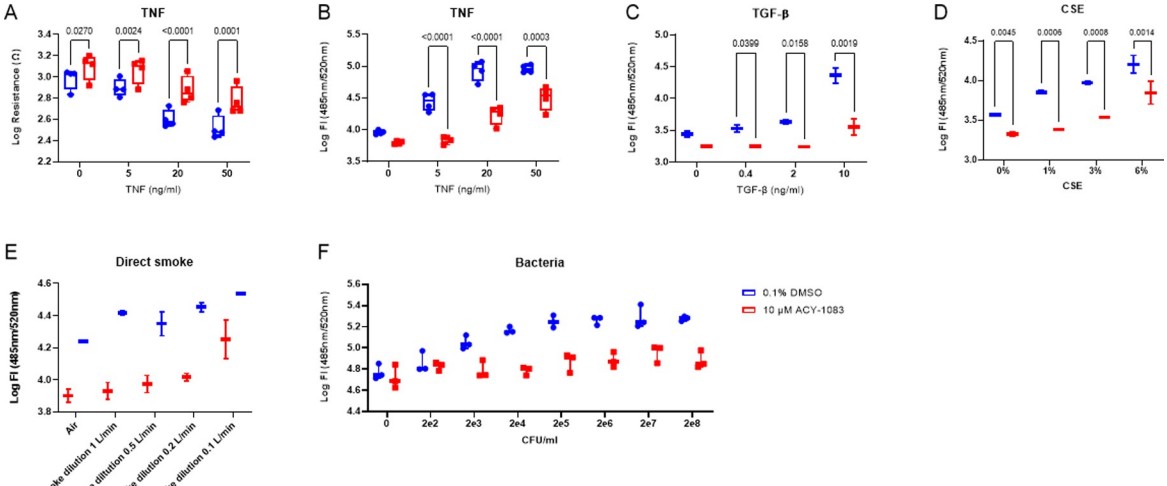

**Fig 2. ACY-1083 improves the epithelial integrity in fully differentiated HBEC ALI cultures challenged with different barrier disruptors.** (A) Resistance measurements after 7 days of TNF-challenge (0–50 ng/ml), or (B) FITC-Dextran measurement after 8 days of TNF challenge (0–50 ng/ml), with or without 10 µM ACY-1083. Box plot with dots representing the average of three replicates for each of 4 COPD donors and whiskers showing min and max. (C) FITC-Dextran measurement after 2 days of TGF-β-challenge (0–10 ng/ml) on cultures pretreated for 3 day with 10 µM ACY-1083 or vehicle. Box plot with dots representing the average of three replicates for each of 2 COPD donors and whiskers showing min and max. (D) FITC-Dextran measurement after 48 hours CSE-challenge (0–6%) on cultures pretreated with 10 µM ACY-1083 or vehicle for 3 days. Box plot with dots representing the average of three replicates for each of 2 COPD donors and whiskers showing min and max. (E) 4 exposures to whole cigarette smoke during 2 days. Treatment with 10 µM ACY-1083 or vehicle started after the first WCS exposure and permeability was measured 20 hours post last exposure. Box plot with dots representing two replicates of 1 COPD donor and whiskers showing min and max. (F) 2 hour incubation of serial diluted *Haemophilus influenzae* followed by wash and permeability measurement at 18 hours. Cultures were pre-treated with 10 µM ACY-1083 or vehicle for 8 days. Representative graph of 1 COPD donor run at three different occasions. Box plot with dots representing three replicates of 1 COPD donor. Statistical analysis was performed using two-way ANOVA with Sidak´s multiple comparisons test.

TGF-β increased permeability and ACY-1083 decreased it at all concentrations (0.4, 2 and 10 ng/ml) (Fig 2C). The same pattern was seen for CSE where permeability concentration dependent and treatment with ACY-1083 revealed a reduction in permeability at all concentrations (1, 3 and 6%) (Fig 2D). For WCS, the window was even greater and ACY-1083 could at all concentrations used except for the highest (1, 0.5 and 0.2 l/min dilution) decrease permeability (Fig 2E). For the bacterial challenge we saw a similar pattern where ACY-1083 reduced paracellular permeability (Fig 2F). In addition, in a pilot experiment to test if ACY-1083 had a bactericidal effect by itself, bacteria suspended in media were incubated for one hour with ACY-1083 revealed no effect on bacterial counts. For all challenges used, ACY-1083 decreased paracellular permeability (Fig 2B–2F). For WCS and bacterial challenge work, only one COPD donor was used for each experiment, the data must thus be considered exploratory despite the clear effect with ACY-1083. These data made us conclude that ACY-1083 reduced the epithelial injury seen after multiple challenges, all relevant to COPD.

## Inflammatory cytokines and mucins were reduced by ACY-1083

To help protecting the lungs from infectious agents and to maintain host defense, epithelial cells are major producers of cytokines and chemokines. To study the inflammatory responses after TNF challenge, several cytokines/chemokines were analyzed from the basolateral supernatants. IL-6, CCL2, and CXCL10 increased dose-dependently by TNF (S3 Fig). Induction of all three cytokines were reduced after simultaneous treatment with ACY-1083 at 5, 20 and 50 ng/ml TNF, indicating that ACY-1083 has a protective effect on the epithelium (Fig 3A–3C).

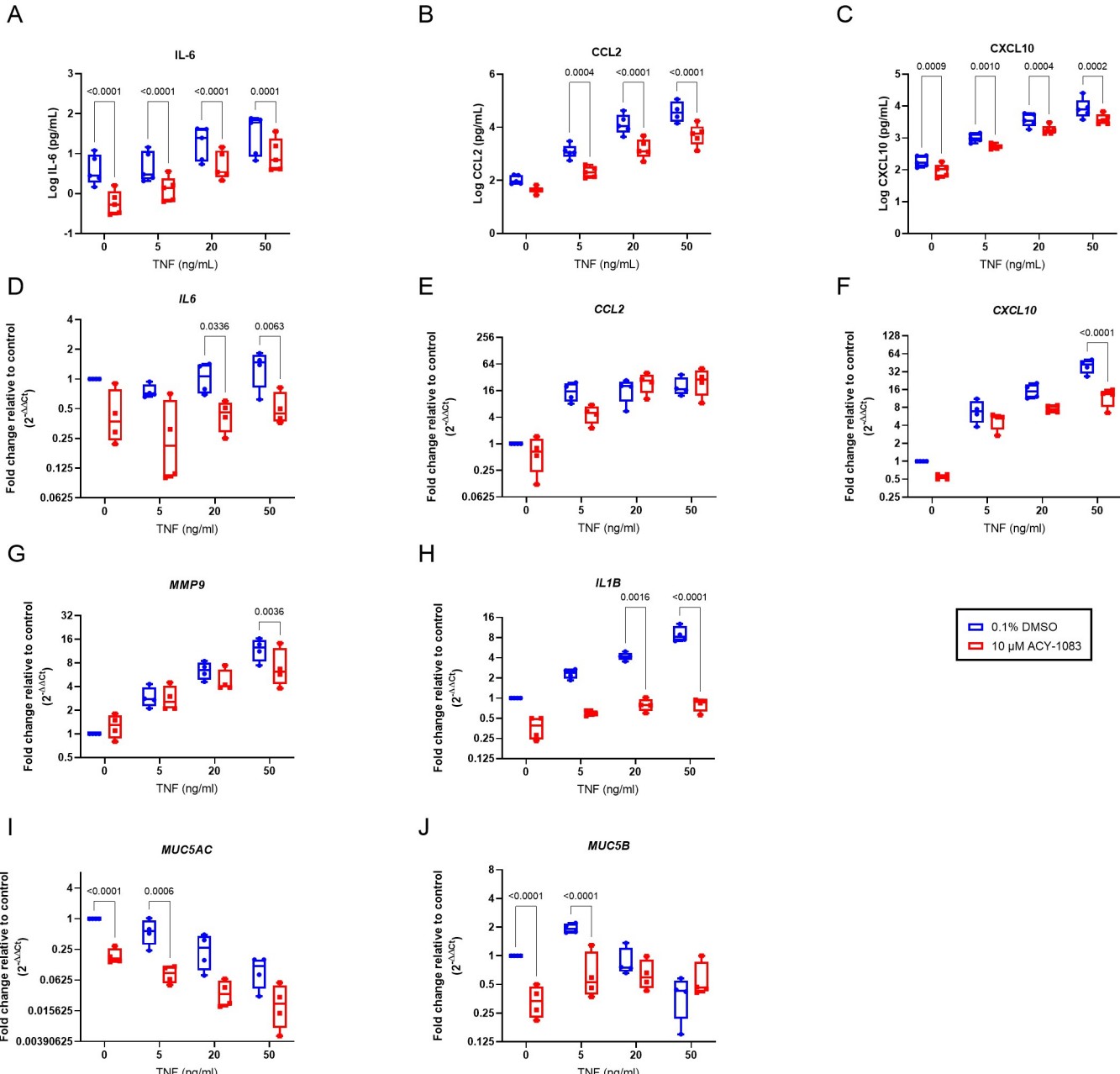

**Fig 3. ACY-1083 reduces pro-inflammatory cytokines and mucins in TNF-challenged HBEC ALI cultures.** Protein concentrations in basolateral supernatants of (A) IL-6, (B) CCL2 and (C) CXCL10 in basolateral supernatants after 7 days of TNF-challenge and ACY-1083/vehicle treatment measured by MSD. Relative mRNA levels of (D) *IL6*, (E) *CCL2*, (F) *CXCL10*, (G) *MMP9*, (H) *IL1B*, (I) *MUC5AC* and (J) *MUC5B* were assessed by RT-PCR from cells lysed after 8 days of TNF-challenge and ACY-1083/vehicle treatment. Box plot with dots representing the average of three replicates for each of 4 COPD donors and whiskers showing min and max. Statistical analysis was performed using two-way ANOVA with Sidak´s multiple comparisons test.

Upon visual examination in a microscope, the wells treated with ACY-1083 looked more similar to the non-TNF control with a brighter apical layer. We therefore proceeded with RNA analysis of the cell cultures to further investigate mucins as well as different cytokines. TNF increased *CCL2*, *CXCL10*, *MMP9*, *IL1B* and reduced the mucin *MUC5AC* (S3 Fig). Treatment with ACY-1083 significantly reduced the transcripts of *IL6*, *CXCL10*, *MMP9* and *IL1B* (Fig 3D

and 3F–3H) at high concentrations of TNF. ACY-1083 also reduced baseline production of *MUC5AC* and *MUC5B* in unchallenged cells (Fig 3I and 3J) as well at 5 ng/ml TNF but had no significant effect at higher concentrations of TNF.

## ACY-1083 protects from morphological changes during epithelial injury

After having established that there was significant inflammation present after TNF-challenge, in addition to the increased permeability, we also wanted to investigate whether the damage of the epithelium had been substantial enough to give morphological changes. Cell cultures were fixed for immunohistology and sections were stained with hematoxylin as well as AB/PAS. Visual assessment of the morphology of no-TNF DMSO-treated cultures showed a pseudostratified epithelium with basal cells at the membrane and a mix of ciliated and goblet cells on the apical side (Fig 4A, left panel). TNF-treated cultures displayed in a dose-dependent fashion, a damaged epithelium with holes, mucus spread within the cell layers, an uneven apical side with patchy spots of ciliated cells and an unorganized basal cell layer (Fig 4A, left panel). Therefore, we concluded that the challenge was severe enough to cause visual epithelial injury. Examining cell cultures treated with ACY-1083 during TNF-challenge showed less damage, intact goblet cells, an even apical surface and a more organized epithelium (Fig 4A, right panel). Sections from all four donors are shown in S4 Fig.

To be able to get an unbiased measurement of the morphological changes seen by eye, we developed a pipeline that was able to discriminate between goblet cells stained with AB, and other cells. After TNF challenge (20 ng/ml), the number of goblet cells in relation to the total number of cells in the sections were decreased as compared to the no-TNF DMSO control (p = 0.0111) (Fig 4B). Interestingly, AB staining was observed in between the cells in TNF-challenged epithelium, and quantification also showed significant increase in this intraepithelial and intercellular areas (p<0.0001) (Fig 4C). After treatment with ACY-1083, the percentage of goblet cells increased but did not differ significantly from the no-TNF DMSO control or the TNF DMSO culture (Fig 4B), and the intracellular AB area was reduced to the no-TNF

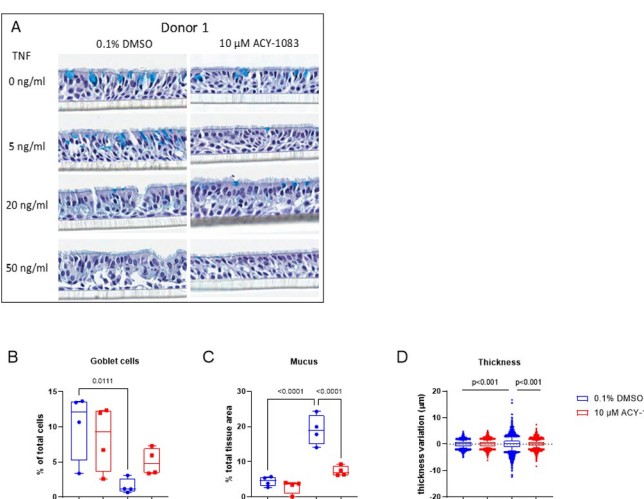

**Fig 4. ACY-1083 protects COPD HBEC ALI cultures from TNF-induced damage.** (A) AB/PAS staining of sections of ALI cultures show pseudostratified epithelium with goblet cells stained in strong blue. Quantification of (B) goblet cell numbers, (C) intercellular mucus within the epithelium and (D) thickness variation. Box plot with whiskers representing min and max for B and C, 10–90 percentile for D, for each of 4 COPD donors. Statistical analysis was performed using one-way ANOVA with Dunnett´s multiple comparisons test.

level (p<0.0001) (Fig 4C). No difference in goblet cell percentage was detected between samples treated with only ACY-1083 or no-TNF DMSO (Fig 4B). Looking at individual donors, three out of four donors had a reduced percentage of goblet cells with TNF DMSO stimulation, which was reverted by ACY-1083 treatment, whereas one donor did not show this pattern (S5 Fig). We therefore concluded that treatment with ACY-1083 partly prevents the change in goblet cell count and protects cells from the intraepithelial and intracellular AB staining seen during TNF challenge, most likely due to a reduced damage of the epithelium.

The apical surface was another striking observation from the IHC sections, where ACY-1083-treated cell cultures appeared more even and with a maintained tight epithelium. In a similar fashion, we developed a pipeline that detected a drop in height at the apical surface and logged the difference in total height change. Comparing DMSO with TNF challenge in four donors, there was an increase in thickness variation (p<0.001). After treatment with ACY-1083 in TNF-challenged wells, the variance in height was smaller in the ACY-1083-treated cell cultures (p<0.001) indicating a rescue of the uneven epithelial surface caused by TNF challenge (Fig 4D). Despite being exploratory, this could also clearly be seen in HBEC ALI cultures stained with occludin and imaged with confocal microscopy. The epithelial cell layer looked more even and tighter after ACY-1083 treatment, but was not quantified so can only be considered as descriptive (S6 Fig).

## Tight junction protein levels are unchanged in membrane fractions after ACY-1083 treatment

After establishing that there were quantifiable changes of the morphology, we analysed the levels of E-cadherin, β-catenin and Occludin in membrane protein extractions from four COPD donors from ALI cell cultures treated with TNF and ACY-1083. TNF treatment did not seem to affect the membrane levels noteworthy of these three proteins, and at 10 μM ACY-1083 the expression was not significantly higher even though a trend could be observed (Fig 5A–5C).

## ACY-1083 protects mucociliary clearance and cellular velocity during TNF challenge

Since treatment with ACY-1083 protected against TNF-induced morphological changes, we next studied cilia function. At day ten, fluorescent beads were added on the apical side and tracked with confocal microscopy and the velocity of the beads were analysed and used as a

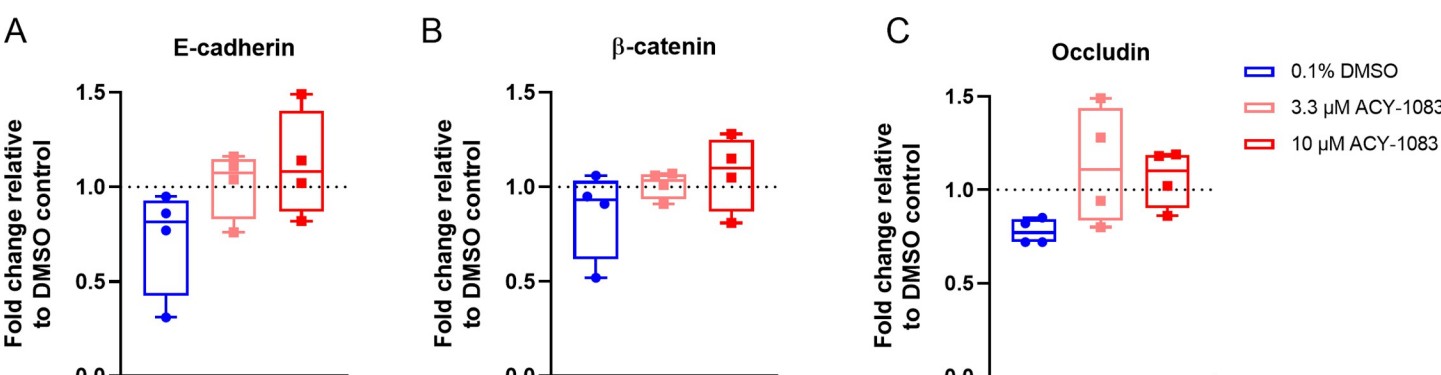

**Fig 5. ACY-1083 does not increase tight-junction proteins in TNF-challenged HBEC ALI cultures.** Western blot analysis of membrane proteins normalized to DMSO control without TNF (dotted line), (A) E-cadherin (B) β-catenin (C) Occludin. Box plot with dots representing 4 COPD donors and whiskers showing min and max. Statistical analysis was performed using one-way ANOVA with Dunnett's multiple comparisons test.

measure of mucociliary clearance (Fig 6A). Quantification showed that TNF drastically reduced the bead movement as compared to the no-TNF control (Fig 6B). ACY-1083 treatment significantly protected from reduction in speed observed after TNF challenge (p = 0.0026) (Fig 6B).

We further evaluated the effect of TNF on the airway epithelial plasticity phenotypes by measuring the barrier function, CBF, and cellular velocity. As previously seen, TNF decreased TEER (p = 0.0023) and ACY-1083 reversed this decrease (p = 0.0187) (Fig 7A). As for CBF, there might be a slight increase after TNF treatment but ACY-1083 had no effect on this measurement (Fig 7B). TNF increased cellular velocity (p = 0.0241) and interestingly ACY-1083 could counteract this increase (0.0411) bringing it down to unchallenged levels (Fig 7C).

## Discussion

Airway epithelial barrier dysfunction, such as increased permeability and morphological alterations, has been demonstrated to be present early in COPD progression and their lungs show increasing destruction and inflammation. In this paper we have sought to investigate the ability to protect and restore the airway epithelial barrier function by using ACY-1083, an HDAC6 inhibitor with superior selectivity over other HDACs, in COPD preclinical model systems. We have used disease-relevant stimuli, such as TNF, TGF-β, CS and bacterial challenge on COPD primary epithelial cells, to establish *in vitro* challenge systems displaying signs of destruction of the epithelial barrier. ACY-1083 dramatically protects the airway epithelial structures and functions *in vitro* which indicate a potential to protect and restore the epithelial barrier, and thus disrupt disease progression, in COPD.

Tight junctions and adherence junctions are forming the barrier that is important for host defense and maintenance of normal epithelial functions. A first sign of an affected epithelium is that the barrier breaks and starts leaking with downstream effects being remodeling and inflammation. Epithelial injury has been discussed as a driver of disease progression and

A

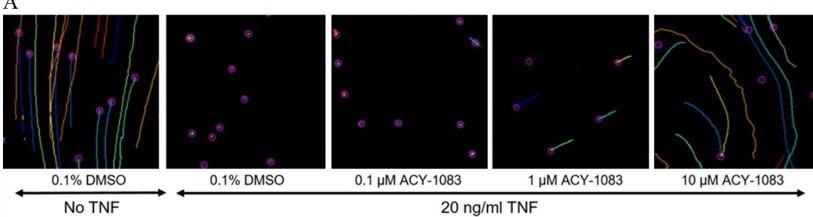

B

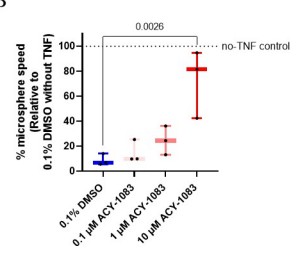

**Fig 6. ACY-1083 maintains mucociliary clearance during TNF-challenge.** Mucociliary transport rates were determined by tracking fluorescent microspheres placed on top of cell layer of unchallenged or TNF-challenged HBEC ALI cultures with or without ACY-1083 treatment. (A) Images of tracked beads in the ImageJ software with the Fiji plugin Tracking. Lines show movement of beads. (B) Microsphere speeds determined from 3 COPD donors and three fields per well. Box plot with dots representing 3 COPD donors and whiskers showing min and max. Data was normalized to the DMSO-control without TNF for each donor (set to 100%). Statistical analysis was performed using one-way ANOVA with Dunnett´s multiple comparisons test.

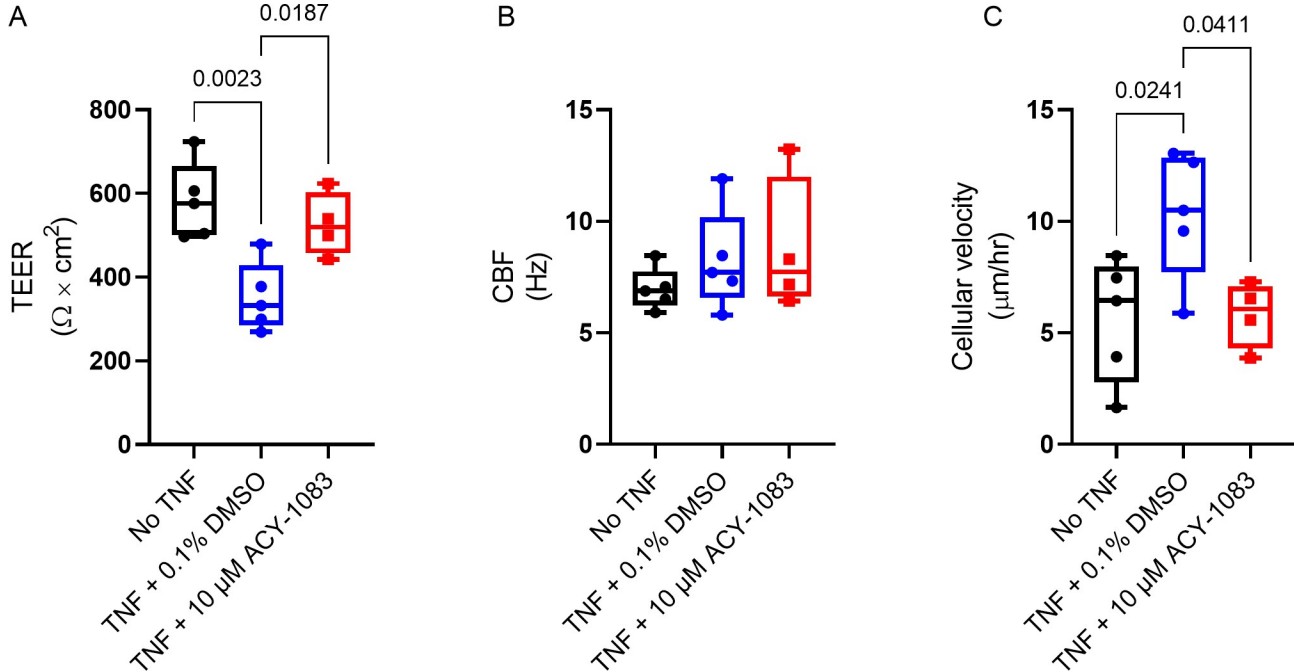

**Fig 7. ACY-1083 preserves epithelial plasticity phenotypes induced by TNF.** Epithelial plasticity phenotypes were determined by measuring (A) TEER, (B) CBF and (C) Cellular velocity of unchallenged and TNF-challenged healthy HBEC at ALI cultures with and without ACY-1083 treatment. Box plot with dots representing the average of triplicates for 3 healthy HBEC donors (unchallenged and TNF + 0.1% DMSO) and the average of triplicates for 2 donors (TNF + ACY-1083) and whiskers showing min and max. Statistical analysis was performed using one-way ANOVA with Dunnett´s multiple comparisons test.

defects in repair mechanisms in lung diseases have been established [61]. Based on this, we set out to study barrier dysfunction in our cell models of COPD-relevant epithelial injury. By challenging ALI cell cultures derived from COPD BECs with TNF, we observed a dose-dependent increase in destruction as demonstrated by changes in cell culture morphology (IHC), increased permeability, inflammatory cytokines, and effects on ciliary function. In these models, ACY-1083 maintained the epithelial integrity at almost unchallenged levels which made us speculate that ACY-1083 treatment must have effects beyond permeability. When analyzing cytokine and protein responses there was a clear downregulation of inflammation as well as mucin production, with cell morphology being markedly protected from damage. To try to understand the exact mechanism by which ACY-1083 is having its protective effects, we analyzed membrane fractions of the ALI cell cultures to see if key molecules in the adherence and tight junctions were altered by ACY-1083 treatment. Surprisingly, the effects on E-cadherin, β-catenin and occludin were modest. Exploratory imaging of occludin staining reveals what looks like a more localized and sharp image as compared to the TNF-treated control, but due to method limitations no quantification was performed to establish this. There have however been reports that β-catenin and other tight junction markers have been increased at the membrane after HDAC6 inhibition [35, 37, 62]. Even though there might be a role for HDAC6 in tight junctions and epithelial permeability, and β-catenin being one of its substrates, we cannot conclusively determine the mechanism for the epithelial protective features of ACY-1083 that we are observing.

A consequence of epithelial injury can be remodeling and EMT, a feature that is characterized by loss of epithelial markers such as E-cadherin and upregulation of mesenchymal markers. This could lead to loss of ciliated cells which in turn would affect the clearance of mucus

and pathogens. It is established that COPD patients have increased remodeling and production of mucins, shorter cilia and alterations in cilia-related genes [28]. The ciliated cells in the TNF-challenged ALI cell cultures looked unaffected, but when analysing bead movement as a model of mucociliary clearance capacity, ACY-1083 significantly increased the essentially eradicated ability to displace mucus/beads observed with TNF challenge. In line with these findings, a few papers have shown protective effects of HDAC6 inhibition on cilia destruction after challenges both *in vitro* and *in vivo* [33, 34]. However, we believe that the effects that we are observing are likely due to reduced damage and intact cells rather than a direct effect on the cilia, despite the described function of HDAC6 in regulating the cilia cell cycle [63]. This is also supported by the fact that CBF was not changed after TNF challenge and ACY-1083 did not alter this. Another outcome of injury and loss of epithelial features such as E-cadherin is that cells start moving and become less tightly attached to one another. When measuring cell velocity in our models, we observed that ACY-1083 completely normalized the movement triggered by the TNF challenge. If this is due to the fact that E-cadherin and other tight junction markers were trending towards an increase in the membrane fractions can only be speculated. Potentially the observed reduced cytokine secretion which could impact the overall inflammatory milieu in the media, is impacting this. In conclusion, we believe that in this model of epithelial injury, we had no effect on cilias *per se* and no potential effect by ACY-1083 could be observed, and rather propose that epithelial features and morphology were protected from injury after treatment with ACY-1083.

Establishing the role for a target by means of pharmacological evaluation is indeed sensitive as there may be unknown secondary target effects. We have evaluated the secondary pharmacology binding and enzymatic panels of ACY-1083 in both a panel of HDAC isoforms and in secondary pharmacology safety panels with only 4 hits below 10 μM (HDAC5, HDAC7, COX2 and DAT). Moreover, we have tried to make sure we do not use higher concentrations than needed with regards to full inhibition at HDAC6. A 10-fold $IC_{50}$ (i.e. nearby full inhibition) from a cell assay would indicate full HDAC6 blockade at 0.3–1 μM [49], but more complicated cell systems including membranes and equilibration of two compartments (basolateral and apical) as in the ALI system, require approximately 10-fold higher concentrations from previous experience. Moreover, the ACY-1083 concentrations in this work (often 10 μM) are the theoretical final basolateral concentrations after addition of the drug to the basolateral compartment. However, we conducted a concentration evaluation of ACY-1083 and observed that approximately 40% is remaining after a simple spike test, ie compound does get lost in the complication of the system (to plastic and media components). Taken together, we are not expecting any secondary pharmacological effects at approximately 3–10 μM concentrations of ACY-1083 from the secondary pharmacology targets tested. Safety aspects of HDAC6 inhibition locally in the lung and literature on HDAC6 KO mice suggest this is a safe target [64, 65] and pharmacological intervention where the catalytic effects are inhibited should be no different.

As with all *in vitro* work, our study has limitations. We have a limited number of donors for our experiments, and some are also merely to be considered exploratory. Despite this fact, our results are very convincing and ACY-1083 succeed in protecting the epithelium from injury. The underlying mechanism for this is also something that we have tried to address by having a broad and diverse set of functional assays, but it is unclear to us exactly how ACY-1083 has this protecting effect. To better understand this we believe that more molecular approaches may be helpful, such as next generation sequencing, as well as proteomics, where pathways and acetylation of proteins can be investigated. Since some aspects of our results could not confirm published findings, eg the cilia involvement described for other HDAC6 inhibitors, we also suggest a more targeted approach eg direct knock-out of HDAC6 in relevant human

primary cells to validate that this has the expected effect. Lastly, this study does not involve any *in vivo* work. As with other targets, mechanistic understanding is sometimes easier to establish *in vivo*, as are the effects of an inhibitor such as ACY-1083, and this could be addressed in future work.

In conclusion, our study suggests that ACY-1083 has the potential to protect the lung epithelium from damage caused by external toxins such as cigarette smoke and pollution, as well as reduce the induction of cytokines induced by such challenges. This implies promise for future treatment opportunities in respiratory disease, which may even be disease modifying, such as in early COPD. As the airway epithelium is "at the surface", local therapeutic options by means of inhalation to restore airway epithelial integrity and function with minor systemic exposure further increase the attractiveness of intervening at the barrier.

## Supporting information

**S1 Fig. Structures of compounds used.**
(TIF)

**S2 Fig. Different barrier disruptors reduce the epithelial integrity in fully differentiated HBEC ALI cultures.** (A) Resistance measurements after 7 days of TNF-challenge (0–50 ng/ml). Box plot with dots representing the average of three replicates for each of 4 COPD donors and whiskers showing min and max. (B) FITC-Dextran measurement after 8 days of TNF challenge (0–50 ng/ml). Box plot with dots representing the average of three replicates for each of 4 COPD donors and whiskers showing min and max. (C) FITC-Dextran measurement after 2 days of TGF-β-challenge (0–10 ng/ml). Box plot with dots representing the average of three replicates for each of 2 COPD donors and whiskers showing min and max. (D) FITC-Dextran measurement after 48 hours CSE-challenge (0–6%). Box plot with dots representing the average of three replicates for each of 2 COPD donors and whiskers showing min and max. Statistical analysis was performed using one-way ANOVA with Dunnett´s multiple comparisons test.
(TIF)

**S3 Fig. TNF increases pro-inflammatory cytokines and decreases mucins in HBEC ALI cultures.** Protein concentrations in basolateral supernatants of (A) IL-6, (B) CCL2 and (C) CXCL10 in basolateral supernatants after 7 days of TNF-challenge measured by MSD. Relative mRNA levels of (D) *IL6*, (E) *CCL2*, (F) *CXCL10*, (G) *MMP9*, (H) *IL1B*, (I) *MUC5AC* and (J) *MUC5B* were assessed by RT-PCR from cells lysed after 8 days of TNF-challenge. Box plot with dots representing the average of three replicates for each of 4 COPD donors and whiskers showing min and max. Statistical analysis was performed using one-way ANOVA with Dunnett´s multiple comparisons test.
(TIF)

**S4 Fig. ACY-1083 protects COPD HBEC ALI cultures from TNF-induced damage.** AB/PAS staining of sections of ALI cultures from 4 COPD donors challenged with different concentrations of TNF (0–50 ng/ml) with and without 10 μM ACY-1083.
(TIF)

**S5 Fig. Goblet cells are protected from TNF challenge by ACY-1083 in HBEC ALI cultures in three out of four donors.** Quantification of goblet cell numbers in HBEC ALI sections stained with AB/PAS from 4 COPD donors.
(TIF)

**S6 Fig. Occludin staining using confocal microscopy on ALI cell cultures treated with ACY-1083 during TNF challenge.** Occludin (in red) and nuclei (in blue) staining of HBEC

ALI cultures from 1 COPD donor challenged with 20 ng/ml TNF. Images showing intersection of cell layer treated with A) vehicle or B) 10 μM ACY-1083. Occludin staining from the same sections as in A-B, C) showing vehicle and D) 10 μM ACY-1083.
(TIF)

**S1 Raw images.**
(PDF)

**S2 Raw images.**
(PDF)

# Acknowledgments

We would like to thank Johns Hopkins University School of Medicine Microscope Facility for providing access to 3i Marianis Spinning Disk Confocal.

We thank Petter Svanberg for the bioanalys of ACY-1083, Thomas Marlow for help with statistics regarding the IHC quantifications and Ken Grime and Linda Yrlid for reading the manuscript.

# Author Contributions

**Conceptualization:** Jenny Horndahl, Annika Åstrand, Venkataramana K. Sidhaye, Mia Collins.

**Data curation:** Jenny Horndahl, Rebecka Svärd, Pia Berntsson, Cecilia Wingren, Jingjing Li, Suado M. Abdillahi, Baishakhi Ghosh, Erin Capodanno, Justin Chan, Lena Ripa, Venkataramana K. Sidhaye, Mia Collins.

**Formal analysis:** Jenny Horndahl, Rebecka Svärd, Pia Berntsson, Cecilia Wingren, Jingjing Li, Suado M. Abdillahi, Baishakhi Ghosh, Erin Capodanno, Justin Chan, Lena Ripa, Venkataramana K. Sidhaye, Mia Collins.

**Investigation:** Jenny Horndahl, Lena Ripa, Annika Åstrand, Venkataramana K. Sidhaye, Mia Collins.

**Methodology:** Jenny Horndahl, Rebecka Svärd, Pia Berntsson, Cecilia Wingren, Jingjing Li, Suado M. Abdillahi, Baishakhi Ghosh, Venkataramana K. Sidhaye, Mia Collins.

**Project administration:** Annika Åstrand, Mia Collins.

**Resources:** Annika Åstrand.

**Supervision:** Annika Åstrand, Mia Collins.

**Validation:** Annika Åstrand, Mia Collins.

**Visualization:** Lena Ripa, Annika Åstrand, Venkataramana K. Sidhaye, Mia Collins.

**Writing – original draft:** Jenny Horndahl, Rebecka Svärd, Pia Berntsson, Cecilia Wingren, Jingjing Li, Suado M. Abdillahi, Baishakhi Ghosh, Erin Capodanno, Justin Chan, Lena Ripa, Annika Åstrand, Venkataramana K. Sidhaye, Mia Collins.

**Writing – review & editing:** Jenny Horndahl, Rebecka Svärd, Pia Berntsson, Cecilia Wingren, Jingjing Li, Suado M. Abdillahi, Baishakhi Ghosh, Erin Capodanno, Justin Chan, Lena Ripa, Annika Åstrand, Venkataramana K. Sidhaye, Mia Collins.

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
