## [Decision Letter · Decision Letter 0]

2 Jun 2022

PONE-D-22-07956HDAC6 inhibitor ACY-1083 shows lung epithelial protective features in COPDPLOS ONE

Dear Dr. Collins,

Thank you for submitting your manuscript to PLOS ONE. After careful consideration, we feel that it has merit but does not fully meet PLOS ONE’s publication criteria as it currently stands. Therefore, we invite you to submit a revised version of the manuscript that addresses the points raised during the review process.

The authors should provide following information to address the Reviewers’ concerns:

1. include additional details related to the donor sample including gender, age, COPD severity, lung function.

2. provide details related to the power analysis used to determine the sample size,  the number of donors per experiment, type of the statistical analyses

3. include the loading controls for the western blotting assays

4. provide additional details on the methods

We look forward to receiving your revised manuscript.

Kind regards,

Yulia Komarova

Academic Editor

PLOS ONE

Journal Requirements:

I have read the journal's policy and the authors of this manuscript have the following competing interests: AstraZeneca employees may hold shares in the company. 

Reviewers' comments:

Reviewer's Responses to Questions

**Comments to the Author**

1. Is the manuscript technically sound, and do the data support the conclusions?

Reviewer #1: Yes

Reviewer #2: Partly

2. Has the statistical analysis been performed appropriately and rigorously? 

Reviewer #1: Yes

Reviewer #2: Yes

3. Have the authors made all data underlying the findings in their manuscript fully available?

Reviewer #1: Yes

Reviewer #2: Yes

4. Is the manuscript presented in an intelligible fashion and written in standard English?

Reviewer #1: Yes

Reviewer #2: Yes

5. Review Comments to the Author

Reviewer #1: The manuscript by Horndahl and colleagues demonstrated that ACY-1083, a selective HDC6 inhibitor, showed protection of the respiratory epithelium treated with TNFa, TGFb, cigarette smoke and Haemophilus infection in air liquid interface cultured COPD bronchial epithelial cells. Thus, an HDAC6 inhibitor has potentials to restore structure or function impaired in COPD. This study is organised very well. The strengths of this study are 1)used air liquid interface cultured epithelium rather than submerged monolayer cells, 2)full characterization of ACY-1083 (primary and secondary pharmacology). 3)used several relevant stumulats 4)used many readout including TEER, clearance, cilia beat etc. Findings are interesting and convincing, and the quality of experiments is at high standard.

However, I have several concerns and questions, and hope that authors are able to address these issues.

1) Major concern is a lack of patient information. What kind of COPD patients are used? Please give more details of donors, including gender, age, COPD severity, lung function.

2) ALI culture should be treated as an individual, and I saw some experiments were conducted using only one donor. The data from one donor is not representative. Please increased number of donors, for example bacteria work.

3) I still feel 10µM is high for the work. I understand authored mentioned that more complicated cell systems including membranes and equilibration of two compartments (basolateral and apical) as in the ALI system, require approximately 10-fold higher concentrations from previous experience. What kind of experience? How can you generalise your finding (x10-fold")? Please check exposure level in ALI epithelium after treatment.

4) Authors used only COPD ALI, but to understand the finding is specific to COPD or not, a small study with healthy ALI (if possible asthma ALI) is suggested.

Reviewer #2: 1. Reviewer report

2022.05.11

Recommendation:

Accept with moderate changes

Comments to the author:

Manuscript number: PONE-D-22-07956

Title: HDAC6 inhibitor ACY-1083 shows lung epithelial protective features in COPD

General comment and summary:

The manuscript presented by Horndahl et al addresses the role of HDAC6 in airway epithelial damage in the context of COPD. The authors motivate for the use of ACY-1083, an inhibitor with high selectivity for HDAC6, and characterized its effects on epithelial function including epithelial disruption, cytokine production, remodeling, mucociliary clearance and cell characteristics. I think the manuscript is well-presented but could be improved by some mild language editing and the moderate edits I suggest below.

Main comments:

1. General commentary, the figure quality needs to be improved as it was exceptionally difficult to assess the manuscript.

2. L144- ‘…five donors of primary healthy HBECs…’ why did you decide on 5 donors per group? Was any a priori sample calculation conducted? Please could you clarify the number of donors per experiment using n’s in each figure legend.

3. L249- You mention the antibodies that were used in these studies. However, concerningly, you do not mention or show any membrane control such as GAPDH. Was there a reason for this? Could you also explain the non-specific bands that appear on the blots (below 50 kDa)?

4. L343- Is there any information regarding the classification of these donor cells? What COPD GOLD stage were these donors?

5. Whilst the epithelial ex vivo assays utilized in this manuscript are well-conducted, what translational value do these data have given the absence of an in vivo model?

6. L375- Nicely planned experiments with relevant perturbations utilized in the probing of this model. How was the MOI of H. Influenzae chosen? Is this clinically relevant?

7. L402- ‘…permeability measurement at 18 hours.’ Why was 18h the time point chosen for permeability measurements. Please clarify this in the manuscript.

8. L480- You would certainly expect a change in these epithelial-associated proteins. I would recommend checking mRNA levels to ensure that the model is producing the desired molecular changes.

9. L495- ‘At day ten, fluorescent beads were added…’ I really enjoyed the thought process behind this assay, however, I have some questions about the translational utility. Were the beads similar to mucous size? Were the beads coated in anything to mimic mucus?

10. L533- ‘…we could mimic the inflammatory milieu…’ I don’t think that this is accurate. The inflammatory milieu consists of immune cells and a number of cytokines not included in these assays. I think this statement should be removed.

11. L564- ‘…cell cultures looked unaffected…’ Why did your cells not express decreased cilia length? This is obviously a crucial proponent of mucociliary clearance.

12. A more detailed description of the limitations and future directions of this study should be included.

Minor comments:

1. Include the molecular weight of the compounds in the figure S1

2. Figure S2. It appears that there are only two measurements in C and D? Is this correct?

3. Figure S5. No statistics have been conducted. Please update this. There also appears to be very little difference in donor 3-could you comment on this?

6. PLOS authors have the option to publish the peer review history of their article (what does this mean?). If published, this will include your full peer review and any attached files.

Reviewer #1: No

Reviewer #2: No

---

## [Author Response · Author response to Decision Letter 0]

14 Jul 2022

I have now provided a file named "Response to Reviewers" where I have addressed all comments made by the editor and Reviewers. 

These are also copied in below. Please let me know if you have any other inquiries. Best wishes and thank you, Mia Collins

ANSWERS TO REVIEWER’S COMMENTS

PONE-D-22-07956

HDAC6 inhibitor ACY-1083 shows lung epithelial protective features in COPD

PLOS ONE

Editors’ summary & main concerns:

Thank you for submitting your manuscript to PLOS ONE. After careful consideration, we feel that it has merit but does not fully meet PLOS ONE’s publication criteria as it currently stands. Therefore, we invite you to submit a revised version of the manuscript that addresses the points raised during the review process.

The authors should provide following information to address the Reviewers’ concerns:

1. include additional details related to the donor sample including gender, age, COPD severity, lung function.

Thank you for this comment. This is relevant information but all our donors are purchased from Lonza and we have very limited information. We have added age, gender and smoking history on these donors in the M&M on line 134 , detailed answer given in the response to Reviewer #1’s first comment below.

2. provide details related to the power analysis used to determine the sample size, the number of donors per experiment, type of the statistical analyses

Thank you for these comments as they are indeed relevant. We believe that we have addressed this in a detailed fashion below (in response to Reviewer #2, second comment)

3. include the loading controls for the western blotting assays

 This has now been clarified below (Reviewer #2, comment 3).

4. provide additional details on the methods

Thank you – We hope that with Reviewer #1 and Reviewer #2’s careful assessments of the manuscript and our responses below, that we have provided sufficient details in the M&M section. 

Comments to the Author

1. Is the manuscript technically sound, and do the data support the conclusions?

Reviewer #1: Yes

Reviewer #2: Partly

2. Has the statistical analysis been performed appropriately and rigorously? 

Reviewer #1: Yes

Reviewer #2: Yes

3. Have the authors made all data underlying the findings in their manuscript fully available?

Reviewer #1: Yes

Reviewer #2: Yes

4. Is the manuscript presented in an intelligible fashion and written in standard English?

Reviewer #1: Yes

Reviewer #2: Yes

5. Review Comments to the Author

Reviewer #1: The manuscript by Horndahl and colleagues demonstrated that ACY-1083, a selective HDC6 inhibitor, showed protection of the respiratory epithelium treated with TNFa, TGFb, cigarette smoke and Haemophilus infection in air liquid interface cultured COPD bronchial epithelial cells. Thus, an HDAC6 inhibitor has potentials to restore structure or function impaired in COPD. This study is organised very well. The strengths of this study are 1)used air liquid interface cultured epithelium rather than submerged monolayer cells, 2)full characterization of ACY-1083 (primary and secondary pharmacology). 3)used several relevant stumulats 4)used many readout including TEER, clearance, cilia beat etc. Findings are interesting and convincing, and the quality of experiments is at high standard.

However, I have several concerns and questions, and hope that authors are able to address these issues.

more details of donors, including gender, age, COPD severity, lung function.

Thank you for this important remark. The donors used, both the COPD and healthy bronchial epithelial cells are originating from epithelial cells purchased from Lonza. We have very limited information about these but we have included age, gender and smoking history for these donors. We have made a new section starting on line 134, now reading:

Patient characteristics from donor cells

Primary COPD human bronchial epithelial cells (HBEC) were purchased from Lonza (Basel, Switzerland). We have used cells from four COPD donors and the information on these donors is limited but include age (years), gender (F/M) and smoking history (yes/no): donor 1 (Cat#00195275, Batch#0000630389, 54, F, yes), donor 2 (Cat#00195275S, Batch 0000370751, 65, F, yes), donor 3 (Cat#00195275S, Batch 0000436083, 59, M, yes), donor 4 (Cat#00195275S, Batch 0000430905, 66, M, yes). Additional five donors of primary healthy HBECs were purchased from Lonza and Epithelix SàRL (Geneva, Switzerland): donor 1 (Cat# hAECB, Batch# AB068001, 71, F, no), donor 2 ( CC-2540, Batch# 0000501935, 56, M, yes), donor 3 (CC-2540S, Batch# 0000619260, 65, F, yes), donor 4 (hAECB, Batch# AB079301, 62, M, no) and donor 5 (hAECB, Batch# #AB077201, 63, M, no).

In doing so, we have also changed figure S5 so that the donors have the same numbering as stated above. This means that it will be donor 1 that does not display the same pattern and response to TNF as the other donors, not donor 3 as in the answer below (Reviewer #2, minor comment 3). 

2) ALI culture should be treated as an individual, and I saw some experiments were conducted using only one donor. The data from one donor is not representative. Please increased number of donors, for example bacteria work.

Thank you for this comment and this observation. It is true that in two experiments we only have one donor, especially with regards to the exploratory experimentations done to explain findings from the planned investigation.

The first one is whole cigarette smoke and the bacteria work. For the bacteria work we repeated the same donor three times and we found tight and convincing data. The whole cigarette smoke experiment has also been repeated several times but due to methodology, it is hard to get a smoke dose where you get sufficient damage to the epithelium, but the cells don’t die. However, with our long experience with ACY-1083, we found it interesting to show the data, despite lack of statistical analysis. Since the bacteria work is part of the test with several stimulations where we see the same effect of ACY-1083, we thought that this work was still worth including, and that it would strengthen the findings, despite being exploratory. We have highlighted “exploratory” in the manuscript, with the sentence at line 403.

For WCS and bacterial challenge work, only one COPD donor was used for each experiment, the data must thus be considered exploratory despite the clear effect with ACY-1083.

The second experiment with n=1 donor is the confocal imaging on the sections from the ALI cultures (S6 Fig). We state that we don’t have the methods to quantify this, but still thought the stainings were interesting to show, as representative for the more even and tight epithelium.

These two experiments are considered exploratory investigations but we are open to removing the data, if preferred by editor/reviewers. Unfortunately we don’t have the possibility to build on this data set and perform more experiments for proper statistical evaluation. We have also here clarified that these data are descriptive and exploratory at line 486: 

Despite being exploratory, this could also clearly be seen in HBEC ALI cultures stained with occludin and imaged with confocal microscopy. The epithelial cell layer looked more even and tighter after ACY-1083 treatment, but was not quantified so can only be considered as descriptive (S6 Fig).

And on line 573 the word “exploratory” has been added and the sentence now reads:

Exploratory imaging of occludin staining reveals what looks like a more localized and sharp image as compared to the TNF-treated control, but due to method limitations no quantification was performed to establish this.

3) I still feel 10µM is high for the work. I understand authored mentioned that more complicated cell systems including membranes and equilibration of two compartments (basolateral and apical) as in the ALI system, require approximately 10-fold higher concentrations from previous experience. What kind of experience? How can you generalise your finding (x10-fold")? Please check exposure level in ALI epithelium after treatment.

This is a great comment and we agree with the reviewers’ concern. We did measure the concentration as stated in the manuscript, and our results indicate that in this case 60% of the compound was lost in the system (line 343). We have long experience with ALI and we have run multiple drug projects using this model. We believe that the ‘loss’ in expected/added drug concentration is due to binding of the drug to the tissue, the plastic in the plate, and proteins in the media that form a surface for the compound to bind to. This is dependent on the physiochemical properties of the drug. We have data indicating that at 10 µM, the only other HDACs that we could possibly be hitting are HDAC5 and 7. As with all pharmacological inhibition, the secondary pharmacology is hard to predict or outrule. ACY-1083 has physiochemical characteristics that should not enrich it inside the cell, but rather equilibrate over the cell membrane and that the measured 4µM after adding 10µM should be what it is also inside the cell. Something that has also been discussed is the use of challenge models in vitro. To demonstrate that you have a drug that can rescue cells from injury over 1-2 weeks, whereas in real life this kind of damage may take years to develop, may also call for higher concentrations of compound. 

4) Authors used only COPD ALI, but to understand the finding is specific to COPD or not, a small study with healthy ALI (if possible asthma ALI) is suggested.

Thank you for this comment and it is a very good suggestion. When we first started using ACY-1083 in epithelial biology, we started with healthy bronchial epithelial cells and we are confident that these findings are not only specific to COPD ALI, but for all kinds of ALI, and cells that form tight junctions. Since this was part of a drug project and we wanted to use as “disease relevant” model systems as possible, we only used COPD ALI for our complete set of experiments. The reason for this is that we discovered that when titrating permeability with different stimulations to validate this model, there were differences in the response to eg TNF. COPD cells were more sensitive and the duration of treatment to achieve a good window for permeability was different versus healthy ALI. Also, our aim was to test the concept and ACY-1083 on COPD lungs, hence we found them more relevant to use for our purposes. But the response to ACY-1083 was always the same, it decreased permeability and the other features that we present. In figure 7 we show cilia characteristics and these experiments are performed on healthy ALI where ACY-1083 shows the expected effect. 

Reviewer #2: 1. Reviewer report

2022.05.11

Recommendation:

Accept with moderate changes

Comments to the author:

Manuscript number: PONE-D-22-07956

Title: HDAC6 inhibitor ACY-1083 shows lung epithelial protective features in COPD

General comment and summary:

The manuscript presented by Horndahl et al addresses the role of HDAC6 in airway epithelial damage in the context of COPD. The authors motivate for the use of ACY-1083, an inhibitor with high selectivity for HDAC6, and characterized its effects on epithelial function including epithelial disruption, cytokine production, remodeling, mucociliary clearance and cell characteristics. I think the manuscript is well-presented but could be improved by some mild language editing and the moderate edits I suggest below.

Main comments:

1. General commentary, the figure quality needs to be improved as it was exceptionally difficult to assess the manuscript.

Thank you for pointing this out. We have corrected this issue. 

2. L144- ‘…five donors of primary healthy HBECs…’ why did you decide on 5 donors per group? Was any a priori sample calculation conducted? Please could you clarify the number of donors per experiment using n’s in each figure legend. 

We have not done any statistical evaluation before the start of the experiments, but have selected sample size based upon our experience of variability in the model. We believe that clinically relevant differences should be able to be seen in a limited number of patients and wells per donor. Even though it would be preferable to have more donors, we see that the results are convincing and statistically significant at this sample size. 

We have added numbers of patients consistently instead of writing with letter and hope that this will faciliate to easily read this. 

3. L249- You mention the antibodies that were used in these studies. However, concerningly, you do not mention or show any membrane control such as GAPDH. Was there a reason for this? Could you also explain the non-specific bands that appear on the blots (below 50 kDa)?

Thank you for pointing this out, it is a mistake on our part that we don’t explain this further. We have used total protein for normalization instead of eg GAPDH. The reason for this is that that it was hard to find a protein that was present in the membrane fractions. 

We have added an explanatory sentence on line 260: 

To correct for variation in loading the total signal intensity of all proteins in each lane were used for normalization. An inter-membrane control sample was loaded to all gels to assure there were no differences due to methodology between the blots. 

As for the unspecific bands, unfortunately we don’t know what these are. 

We will also upload a separate PDF with the total protein staining. 

 4. L343- Is there any information regarding the classification of these donor cells? What COPD GOLD stage were these donors?

Thank you for this important remark. Reviewer 1 has asked the same thing and we write: The donors used, both the COPD and healthy bronchial epithelial cells are originating from cells purchased from Lonza. We have very limited information about these but we have included age, gender and smoking history for these donors on line 134. 

5. Whilst the epithelial ex vivo assays utilized in this manuscript are well-conducted, what translational value do these data have given the absence of an in vivo model? 

This is a very good remark. Since there is no preclinical aspects of this model present, eg immune cells of the lung, fibroblasts and extracellular matrix or blood flow, the translational value cannot be determined. But we have long experience with in vivo models and it is very hard to claim that we have animal models that can mimic human disease. Even with an in vivo model, it wouldn’t give us better confidence that it would work in clinical practice. Our concern with potential mouse models is that HDAC6 lacks the cytoplasm-anchoring motif that makes it enter the nucleus and there also de-acetylates histones.

6. L375- Nicely planned experiments with relevant perturbations utilized in the probing of this model. How was the MOI of H. Influenzae chosen? Is this clinically relevant?

 Thank you for this question. We titrated the dose of bacteria to where we would have a reduction in permeability, as well as having an obvious epithelial injury. The MOI of bacteria was chosen broadly from 2e2 to 2e8, unfortunately we are not sure of the clinical relevance of the doses, but we do think that the assay per se is still relevant. 

7. L402- ‘…permeability measurement at 18 hours.’ Why was 18h the time point chosen for permeability measurements. Please clarify this in the manuscript.

Thank you for this comment. We titrated this carefully and shorter time points gave too small window. 18h were chosen for optimal and practical reasons and we wanted to keep it consistent between assays with eg TNF. A sentence has been added in the M&M at line 170 reading: 

The time point was chosen for optimal window in the assays as well as practical reasons and has been kept consistent in between different stimulations throughout this work. 

to address this uncertainty. As well as a clarifying the time point on line 208, and in the results section at line 381:

18hrs were chosen as time point where we saw a clear window in permeability.

8. L480- You would certainly expect a change in these epithelial-associated proteins. I would recommend checking mRNA levels to ensure that the model is producing the desired molecular changes.

Thank you for this suggestion. We have checked many different genes, including tight junction genes, but we have found it very hard to catch meaningful and reproducible changes. We believe that it is due to the fact that we have a chronic model where the injury happens over time and TNF is added every other day where each donor is responding a bit differently. Over time this builds up to the changes that we present in the manuscript, but as stated, it is hard to determine the exact mechanism/-s and time course for the effects we are seeing with ACY-1083.

9. L495- ‘At day ten, fluorescent beads were added…’ I really enjoyed the thought process behind this assay, however, I have some questions about the translational utility. Were the beads similar to mucous size? Were the beads coated in anything to mimic mucus?

This is a great question. We picked 7µm beads for these experiments. We looked at size of not only mucous (polymeres approximatey 10µm), but also bacteria (1-10µm) and particulate matter (<10µm) which are also transported by cilia. The beads were not coated with anything. 

10. L533- ‘…we could mimic the inflammatory milieu…’ I don’t think that this is accurate. The inflammatory milieu consists of immune cells and a number of cytokines not included in these assays. I think this statement should be removed.

 This is correct and we have removed the statement. The sentence on line 554 now reads: 

We have used disease-relevant stimuli, such as TNF, TGF-β, CS and bacterial challenge on COPD primary epithelial cells, to establish in vitro challenge systems displaying signs of destruction of the epithelial barrier.

11. L564- ‘…cell cultures looked unaffected…’ Why did your cells not express decreased cilia length? This is obviously a crucial proponent of mucociliary clearance.

We believe that with TNF as a stimulus, we don’t affect cilia length per se, but rather affect the level of damage to the epithelium by inflammatory cytokines etc. It seems that this is creating an uneven surface of the apical epithelium and that particles and mucous get stuck and are not moving properly. We speculate that this is the reason for CBF not to be affected by TNF, but possibly the synchronization of the beating is. This is the reason why it is hard to show that ACY-1083 has a direct effect on cilia length, but it should be mentioned that we lack the proper methods to measure this. 

12. A more detailed description of the limitations and future directions of this study should be included.

 Thank you for this suggestion, we have forgotten to raise this important discussion. A section has now been added to the Discussion on line 618. 

As with all in vitro work, our study has limitations. We have a limited number of donors for our experiments, and some are also merely to be considered exploratory. Despite this fact, our results are very convincing and ACY-1083 succeed in protecting the epithelium from injury. The underlying mechanism for this is also something that we have tried to address by having a broad and diverse set of functional assays, but it is unclear to us exactly how ACY-1083 has this protecting effect. To better understand this we believe that more molecular approaches may be helpful, such as next generation sequencing, as well as proteomics, where pathways and acetylation of proteins can be investigated. Since some aspects of our results could not confirm published findings, eg the cilia involvement described for other HDAC6 inhibitors, we also suggest a more targeted approach eg direct knock-out of HDAC6 in relevant human primary cells to validate that this has the expected effect. Lastly, this study does not involve any in vivo work. As with other targets, mechanistic understanding is sometimes easier to establish in vivo, as are the effects of an inhibitor such as ACY-1083, and this could be addressed in future work.

Minor comments:

1. Include the molecular weight of the compounds in the figure S1

This has now been specified on line 111 which now reads: 

ACY-1083 (CAS1708113-43-2, Mw 348.35) is commercially available but was prepared according to a published procedure (50, 51). Tubastatin A (HY-13271, Mw 371.86 HCl salt) and Ricolinostat (HY-16026, Mw 433.5) were obtained from MedChemExpress.

2. Figure S2. It appears that there are only two measurements in C and D? Is this correct?

This is correct, it is two donors, with three individual measurements. 

3. Figure S5. No statistics have been conducted. Please update this. There also appears to be very little difference in donor 3-could you comment on this?

Thank you for this comment. In figure 4B, we have the pooled results from all four donors. To our surprise, there was no significant reduction of goblet cell percentage with ACY-1083 between the no-TNF and the TNF treatment. When looking at the individual donors as shown in S5 Fig, it is clear that donor 3 did not respond to TNF in the same fashion as the other 3 donors. This is likely why we do not reach statistical significance. We can however not calculate statistics on the individual donors in S5 Fig since we for data originating from IHC sections, only have one measurement per donor.

6. PLOS authors have the option to publish the peer review history of their article (what does this mean?). If published, this will include your full peer review and any attached files.

Do you want your identity to be public for this peer review? For information about this choice, including consent withdrawal, please see our Privacy Policy.

Reviewer #1: No

Reviewer #2: No

---

## [Decision Letter · Decision Letter 1]

26 Aug 2022

PONE-D-22-07956R1HDAC6 inhibitor ACY-1083 shows lung epithelial protective features in COPDPLOS ONE

Dear Dr. Collins,

Thank you for submitting your manuscript to PLOS ONE. After careful consideration, we feel that it has merit but does not fully meet PLOS ONE’s publication criteria as it currently stands. Therefore, we invite you to submit a revised version of the manuscript that addresses the points raised during the review process.

The authors should contact Lonza to acquire additional information regarding the donors. This information is usually available upon request. The authors should remove the reference to cilia length as recommended by the Reviewer 2.

We look forward to receiving your revised manuscript.

Kind regards,

Yulia Komarova

Academic Editor

PLOS ONE

Journal Requirements:

Reviewers' comments:

Reviewer's Responses to Questions

**Comments to the Author**

1. If the authors have adequately addressed your comments raised in a previous round of review and you feel that this manuscript is now acceptable for publication, you may indicate that here to bypass the “Comments to the Author” section, enter your conflict of interest statement in the “Confidential to Editor” section, and submit your "Accept" recommendation.

Reviewer #1: All comments have been addressed

Reviewer #2: (No Response)

2. Is the manuscript technically sound, and do the data support the conclusions?

Reviewer #1: Partly

Reviewer #2: Yes

3. Has the statistical analysis been performed appropriately and rigorously? 

Reviewer #1: Yes

Reviewer #2: N/A

4. Have the authors made all data underlying the findings in their manuscript fully available?

Reviewer #1: Yes

Reviewer #2: Yes

5. Is the manuscript presented in an intelligible fashion and written in standard English?

Reviewer #1: Yes

Reviewer #2: Yes

6. Review Comments to the Author

Reviewer #1: Regarding to patient background, the number of donors, authors did not solve the problems fully, but described clearly (exploratory vs. main experiment) or provided best possible information. Also authors use very high concentration of compound for ALI, and from my experiences, the compound has problems in mucosal residency. This is an experimental drug, and all findings are for basic science rather than clinical trial mimic. Therefore, I am satisfied with all comments provided by authors. For future work, I wish authors conduct power calculation and use appropriate number of donors even for ALI work.

Reviewer #2: The author have replied convincingly to the majority of my points. However, I still have a few remaining questions.

(1)Thank you for this important remark. Reviewer 1 has asked the same thing and we

write: The donors used, both the COPD and healthy bronchial epithelial cells are

originating from cells purchased from Lonza. We have very limited information about

these but we have included age, gender and smoking history for these donors on line

134.

I understand that the Lonza bought cells are difficult to get information on. However, the COPD stage is exceptionally important given the different responses you see amongst donors. I would still recommend that this is added.

(2) We believe that with TNF as a stimulus, we don’t affect cilia length per se, but rather

affect the level of damage to the epithelium by inflammatory cytokines etc. It seems

that this is creating an uneven surface of the apical epithelium and that particles and

mucous get stuck and are not moving properly. We speculate that this is the reason

for CBF not to be affected by TNF, but possibly the synchronization of the beating is.

This is the reason why it is hard to show that ACY-1083 has a direct effect on cilia

length, but it should be mentioned that we lack the proper methods to measure this.

I would recommend removing any reference to cilia length in this case.

7. PLOS authors have the option to publish the peer review history of their article (what does this mean?). If published, this will include your full peer review and any attached files.

Reviewer #1: No

Reviewer #2: No

---

## [Author Response · Author response to Decision Letter 1]

12 Sep 2022

Please also see the file named "Response to Reviewers" where the attached patient data can also be found. 

ANSWERS TO REVIEWER’S COMMENTS

PONE-D-22-07956R1

HDAC6 inhibitor ACY-1083 shows lung epithelial protective features in COPD

PLOS ONE

Editors’ summary & main concerns:

Thank you for submitting your manuscript to PLOS ONE. After careful consideration, we feel that it has merit but does not fully meet PLOS ONE’s publication criteria as it currently stands. Therefore, we invite you to submit a revised version of the manuscript that addresses the points raised during the review process.

The authors should contact Lonza to acquire additional information regarding the donors. This information is usually available upon request.

Thank you for this comment, we agree that it is very important information. We have contacted Lonza and we have the table of patient data that they have given us. Unfortunately there is no disease stage data. We have included the table further down in the response to Reviewer #2, first question. 

The authors should remove the reference to cilia length as recommended by the Reviewer 2.

We have changed the manuscript accordingly, please see the response to Reviewer #2, second question for a detailed description of changes. 

Comments to the Author

1. If the authors have adequately addressed your comments raised in a previous round of review and you feel that this manuscript is now acceptable for publication, you may indicate that here to bypass the “Comments to the Author” section, enter your conflict of interest statement in the “Confidential to Editor” section, and submit your "Accept" recommendation.

Reviewer #1: All comments have been addressed

Reviewer #2: (No Response)

2. Is the manuscript technically sound, and do the data support the conclusions?

Reviewer #1: Partly

Reviewer #2: Yes

3. Has the statistical analysis been performed appropriately and rigorously? 

Reviewer #1: Yes

Reviewer #2: N/A

4. Have the authors made all data underlying the findings in their manuscript fully available?

Reviewer #1: Yes

Reviewer #2: Yes

5. Is the manuscript presented in an intelligible fashion and written in standard English?

Reviewer #1: Yes

Reviewer #2: Yes

6. Review Comments to the Author

Reviewer #1: Regarding to patient background, the number of donors, authors did not solve the problems fully, but described clearly (exploratory vs. main experiment) or provided best possible information. Also authors use very high concentration of compound for ALI, and from my experiences, the compound has problems in mucosal residency. This is an experimental drug, and all findings are for basic science rather than clinical trial mimic. Therefore, I am satisfied with all comments provided by authors. For future work, I wish authors conduct power calculation and use appropriate number of donors even for ALI work.

We thank Reviewer 1 for the careful assessment of the manuscript. We will in the future make sure we do power calculations. 

Reviewer #2: The author have replied convincingly to the majority of my points. However, I still have a few remaining questions.

(1)Thank you for this important remark. Reviewer 1 has asked the same thing and we

write: The donors used, both the COPD and healthy bronchial epithelial cells are

originating from cells purchased from Lonza. We have very limited information about

these but we have included age, gender and smoking history for these donors on line

134.

I understand that the Lonza bought cells are difficult to get information on. However, the COPD stage is exceptionally important given the different responses you see amongst donors. I would still recommend that this is added.

We agree that this is important and the GOLD stage might as you are saying give us an explanation as to why the patients are responding differently to the treatments and stimulations. We have again contacted Lonza to ask for this information but they refer us to this table that I have also attached and say that they are not normally able to access information regarding GOLD stages. Had we been using our own patient cohorts or collaborations we would have been able to add this information but unfortunately this is not always the case for commercially available cells. We are sad to say that we have not been able to access the information asked for. 

(2) We believe that with TNF as a stimulus, we don’t affect cilia length per se, but rather

affect the level of damage to the epithelium by inflammatory cytokines etc. It seems

that this is creating an uneven surface of the apical epithelium and that particles and

mucous get stuck and are not moving properly. We speculate that this is the reason

for CBF not to be affected by TNF, but possibly the synchronization of the beating is.

This is the reason why it is hard to show that ACY-1083 has a direct effect on cilia

length, but it should be mentioned that we lack the proper methods to measure this.

I would recommend removing any reference to cilia length in this case.

This is a good point. We have changed two sentences and thereby removed any specific mentioning of cilia length in the discussion. 

“It is established that COPD patients have increased remodeling and production of mucins, shorter cilia and alterations in cilia-related genes” on line 580

and

“In line with these findings, a few papers have shown protective effects of HDAC6 inhibition on cilia destruction after challenges both in vitro and in vivo” on line 584

---

## [Decision Letter · Decision Letter 2]

26 Sep 2022

HDAC6 inhibitor ACY-1083 shows lung epithelial protective features in COPD

PONE-D-22-07956R2

Dear Dr. Collins,

We’re pleased to inform you that your manuscript has been judged scientifically suitable for publication and will be formally accepted for publication once it meets all outstanding technical requirements.

Kind regards,

Yulia Komarova

Academic Editor

PLOS ONE

Additional Editor Comments (optional):

Reviewers' comments:

Reviewer's Responses to Questions

**Comments to the Author**

1. If the authors have adequately addressed your comments raised in a previous round of review and you feel that this manuscript is now acceptable for publication, you may indicate that here to bypass the “Comments to the Author” section, enter your conflict of interest statement in the “Confidential to Editor” section, and submit your "Accept" recommendation.

Reviewer #2: (No Response)

2. Is the manuscript technically sound, and do the data support the conclusions?

Reviewer #2: Yes

3. Has the statistical analysis been performed appropriately and rigorously? 

Reviewer #2: Yes

4. Have the authors made all data underlying the findings in their manuscript fully available?

Reviewer #2: Yes

5. Is the manuscript presented in an intelligible fashion and written in standard English?

Reviewer #2: Yes

6. Review Comments to the Author

Reviewer #2: The authors have attempted to answer the question sufficiently. Whilst it is important to know what GOLD stage the patient samples are obtained from, this does not seem to be within the control of the authors as they state.

7. PLOS authors have the option to publish the peer review history of their article (what does this mean?). If published, this will include your full peer review and any attached files.

Reviewer #2: No

---

## [Editor Report · Acceptance letter]

2 Oct 2022

PONE-D-22-07956R2 

HDAC6 inhibitor ACY-1083 shows lung epithelial protective features in COPD 

Dear Dr. Collins:

I'm pleased to inform you that your manuscript has been deemed suitable for publication in PLOS ONE. Congratulations! Your manuscript is now with our production department. 

Kind regards, 

on behalf of

Dr. Yulia Komarova 

Academic Editor

PLOS ONE